

# Caution ahead: reassessing the functional morphology of the respiratory organs in amphibious snails

Guido I. Prieto

Department of Philosophy I, Ruhr University Bochum, Bochum, Germany

## ABSTRACT

After a long hiatus, interest in the morphology of the respiratory organs in apple snails (Ampullariidae, Caenogastropoda) and its functional and evolutionary bearings is making a comeback. The variability in the morphology of the gill and lung and its link to different lifestyles and patterns of air dependency within the Ampullariidae make research on the morphology of the respiratory organs particularly suitable for approaching the evolution of terrestriality in animals. Additionally, morphology is a valuable source of hypotheses regarding the several functions the ampullariid respiratory organs have besides respiration. However, this is an underexplored field that only recently has incorporated ultrastructural and three-dimension visualization tools and in which more research is much needed, particularly, comparisons between species representing the diversity within the Ampullariidae. In this paper, I examine *Mueck, Deaton & Lee*'s (*2020a*) assessment of the morphology of the gill and lung of *Pomacea maculata* and compare it with earlier and contemporary studies on other ampullariid species. I show that Mueck, Deaton & Lee's paper combines significant morphological misinterpretations, conceptual and terminological mistakes, and crucial literature omissions. I also reinterpret their results and point to the similarities and differences between them and available data on other ampullariids.

## INTRODUCTION

The caenogastropod family Ampullariidae comprises bimodally breathing aquatic snails that have the peculiarity of retaining a gill while having secondarily acquired a lung for air-breathing, which allows them to inhabit poorly oxygenated water bodies (*Berthold, 1991*). However, the dependence on air-breathing varies among the different genera and species in the family. Some species are almost fully aquatic and lay gelatinous eggs underwater, while species in the genera *Pomacea* and *Pila* convergently lay calcareous eggs out of the water and rely more on air-breathing (*Michelson, 1961*; *Hayes et al., 2009*). Comparative and functional morphological studies of the respiratory organs would be particularly valuable for deepening our knowledge of the evolutionary patterns in Ampullariidae. Until recently (*Rodriguez et al., 2019*; *Rodriguez et al., 2021*), however, the respiratory organs had been

Corresponding author
Guido I. Prieto, Guido.Prieto@rub.de

largely overlooked from an evolutionary perspective (see, *e.g.*, *Hayes et al., 2009*), with a few notable exceptions (*Robson, 1922*; *Kretschmann, 1955*; *Berthold, 1991*).

In Ampullariidae, the respiratory organs seem to be involved in manifold vital processes besides gas exchange. The gill helps to generate the water current across the mantle cavity and, in *Pomacea canaliculata*, may have an important role in ionic/osmotic and acid–base regulation and immune defense (*Taylor & Andrews, 1987*; *Rodriguez et al., 2019*). The lung is used as a flotation device (*Bavay, 1873*; *Ramanan, 1903*; *Hylton Scott, 1957*; *McClary, 1965*; *Burky & Burky, 1977*; *Berthold, 1991*) and, in *P. canaliculata*, it is involved in immune defense (*Rodriguez et al., 2018*), hematopoiesis (*Rodriguez et al., 2020*; *Rodriguez et al., 2021*), and storage of urates (*Giraud-Billoud et al., 2008*), which are key components in the antioxidant-defense mechanism during arousal after estivation (*Giraud-Billoud et al., 2013*) and hibernation (*Giraud-Billoud et al., 2018*). It has been reported that the lung of *Pomacea maculata* accumulates calcium carbonate that may buffer the lowering of pH during estivation (*Mueck, Deaton & Lee, 2020b*). Detailed morphological studies can shed light on these functions and point to new ones. For instance, the description of the gill of *P. canaliculata* by *Rodriguez et al. (2019)* has inspired a hypothesis for the mechanism of cadmium incorporation and transport (*Campoy-Diaz et al., 2020*) and encouraged research on the processes of immune response in the gill of this species (*Montanari et al., 2020*).

As relevant as they may be from evolutionary and functional perspectives, detailed morphological studies of the respiratory organs in Ampullariidae are scarce and restricted to a few species (Table 1). Apart from numerous anatomical descriptions based on dissections (*e.g.*, *Hägler, 1923*; *Andrews, 1965*; *Demian, 1965*; *Starmühlner, 1969*; *Berthold, 1991*; *Simone, 2004*), there is a handful of light-microscopical studies and only four ultrastructural accounts, namely those of *Rodriguez et al. (2019)* and *Rodriguez et al. (2021)* on the gill and lung of *P. canaliculata*, *Mueck, Deaton & Lee (2020a)* on the gill and lung of *P. maculata*, and *Mueck, Deaton & Lee (2020b)* on the storage tissue of the lung in the same species. This scarcity of data on the histology and ultrastructure of the respiratory organs is not restricted to Ampullariidae, but holds for gastropods in general and particularly for caenogastropods (see *Rodriguez et al., 2019*; *Mueck, Deaton & Lee, 2020a*).

Considering their seminal character, it is especially necessary that new morphological assessments on the respiratory organs in Ampullariidae be conducted carefully and their results compared with previous literature and put into a functional and evolutionary context. Since *qui tacet consentire videtur*, herein I discuss the problematic morphological descriptions by *Mueck, Deaton & Lee* (*2020a*; hereafter, MDL) and compare them with previous literature on the respiratory organs in Ampullariidae, particularly with the work of *Rodriguez et al. (2018)*, *Rodriguez et al. (2019)* and *Rodriguez et al. (2021)*, which I co-authored. For reasons of space, I focus solely on some central issues. Throughout all the text, emphasis in the quotations is mine.

**Table 1** **Studies on the histology of the respiratory organs of apple snails.** Note the temporal gap between the early, light microscopy studies and the recent studies that include ultrastructural examinations. *Brooks & McGlone (1908)* and *Demian & Yousif (1973)*—who studied the ontogeny of the respiratory organs—as well as *Berthold (1991)*, contain only superficial histological accounts and are thus excluded from the list.

| Author(s) and year | Species | Organ(s) | Method(s) | Visualization of data |
|---|---|---|---|---|
| *Prashad (1925)* | *Pila globosa* | Gill, lung | LM | Micrographs, drawings |
| *Kretschmann (1955)* | *Lanistes lybicus, Ampullaria* (*Pomacea*) *sordida* | Gill, lung | LM | Drawings |
| *Michelson (1955)*[*] | *Ceratodes* (*Marisa*) *cornuarietis* | Gill, lung | LM | ? |
| *Lutfy & Demian (1965)* | *Marisa cornuarietis* | Gill, lung | LM | Drawings |
| *Rodriguez et al. (2018)* | *Pomacea canaliculata* | Lung | LM | Micrographs |
| *Rodriguez et al. (2019)* | *Pomacea canaliculata* | Gill | LM, SEM, TEM, 3D | Micrographs, drawings, interactive 3D models |
| *Mueck, Deaton & Lee (2020a)* | *Pomacea maculata* | Gill, lung | LM, SEM, TEM | Micrographs |
| *Mueck, Deaton & Lee (2020b)* | *Pomacea maculata* | Lung (storage tissue) | LM, SEM, TEM | Micrographs |
| *Rodriguez et al. (2021)* | *Pomacea canaliculata* | Lung | LM, SEM, TEM, 3D | Micrographs, drawings, interactive 3D models |

**Notes.**

Abbreviations: LM, light microscopy; SEM, scanning electron microscopy; TEM, transmission electron microscopy; 3D, three-dimensional reconstructions.

[*]Unpublished Ph.D. thesis cited by *Lutfy & Demian (1965)*.

## CRITIQUE OF MUECK, DEATON & LEE'S MORPHOLOGICAL STUDY

### Histology of the gill

MDL study the microanatomy of the gill with light microscopy (LM) and scanning electron microscopy (SEM). Based on the LM examination, they divide the gill into three 'zones' and enumerate them as I-III from dorsal to ventral. Zone I includes the outer mantle epithelium and the underlying fibromuscular tissue—which are not restricted to the gill, but cover the entire dorsal mantle wall (*Andrews, 1965*; *Lutfy & Demian, 1965*; *Rodriguez et al., 2019*; *Rodriguez et al., 2021*) –, while zones II and III correspond to the basal and middle portions of the gill leaflets, respectively. It should be emphasized that MDL do not examine sections of the distal or apical portion of the gill leaflets with LM or transmission electron microscopy (TEM). Therefore, it is not surprising that they find "[n]o skeletal supporting elements […] inside the filaments" (p. 127), since skeletal rods—when present—are placed along the efferent borders of the gill leaflets in gastropods (*Yonge, 1947*; *Ponder, Lindberg & Ponder, 2019*; see *Rodriguez et al., 2019* for an updated discussion on the presence of skeletal rods in Ampullariidae). Nonetheless, they do study the border of the gill leaflets with SEM.

The problem arises when they attempt to correlate the zones identified under LM with the structures examined under SEM, which are different (Fig. 1A). Roughly, they equate the *base* of the gill leaflets (Fig. 1B) with their free *border* (Fig. 1C). The source of this confusion may be the fact that MDL identify as *cilia* what actually are the *cytoplasms* of the epithelial cells (see their figure 2). From this, they erroneously conclude that "[t]he entire length of the filament […] is ciliated" (p. 129). The low quality and excessive contrast of their figure 2 prevents a straightforward identification of the structures and, ironically, the authors criticize *Eertman (1996)* on similar grounds (p. 129). However, it is clear that

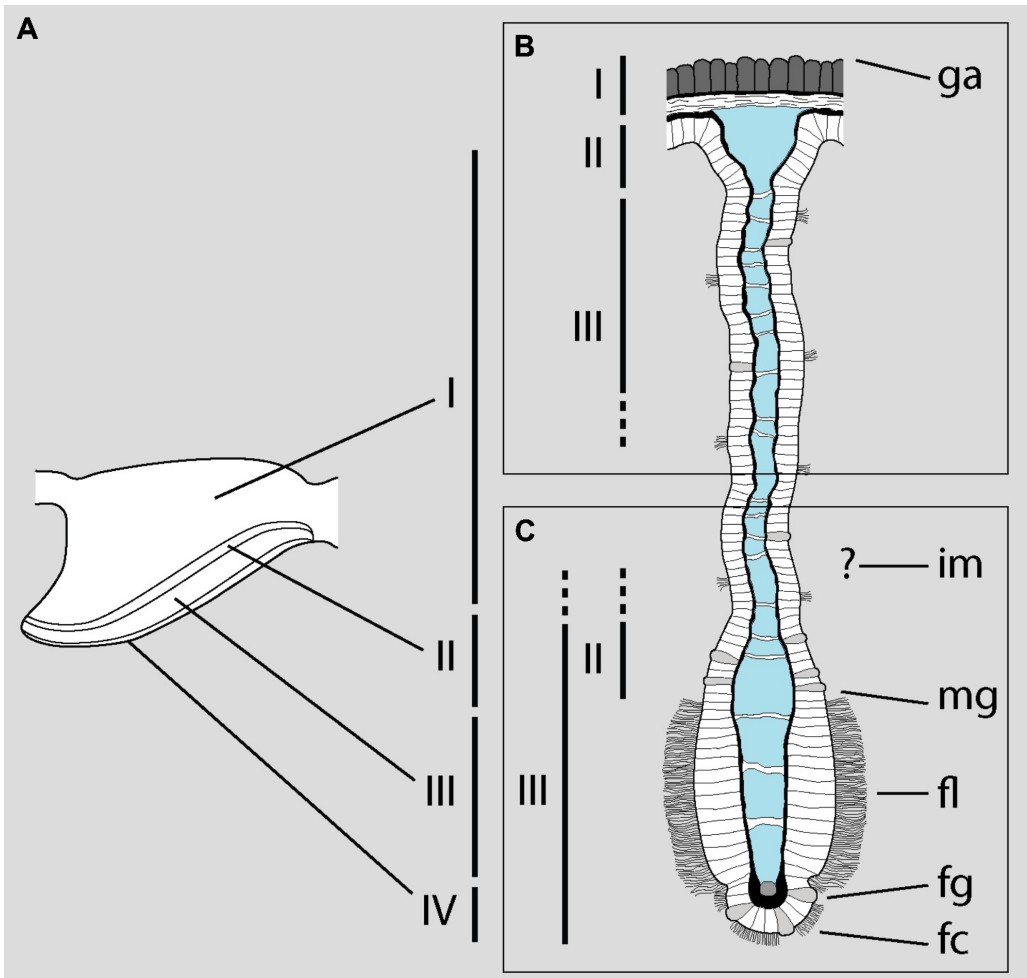

**Figure 1** **Disentangling Mueck, Deaton & Lee's description of the gill of *Pomacea maculata*.** The diagram shows a gill leaflet in lateral view (left) and in section (right). (A) The four regions identified by *Rodriguez et al. (2019)* in the gill leaflets of *P. canaliculata*. (B) Portion of the gill leaflet studied by MDL with LM. (C) Portion of the gill leaflet studied by MDL with SEM. MDL first define zones II and III in (B) and then make them correspond to "the external morphology of the filament from the marginal groove to the inner margin" and "from the inner margin to the apex" (p. 127), respectively (marked as zones II and III in C). The identity of the 'inner margin' is unclear. Note the correlation between basal and apical structures and the overlap of zones II and III in the lower rectangle. Abbreviations as they appear in MDL's paper: fc, frontal cilia; fg, frontal groove; fl, lateral tract of frontal cilia; ga, gill axes [sic]; im, inner margin; mg, marginal groove. Diagram modified from *Rodriguez et al.* (*2019*; DOI: 10.7717/peerj.7342/fig-4).

MDL's "lateral cilia" of "about 25 µM [sic] in length" (p. 127) correspond to the epithelial cells as seen in region I of the gill leaflets in *P. canaliculata*. There, the epithelium is >20 µm in height and is mostly composed of microvilli-bearing and interspersed ciliated cells and secretory cells (*Rodriguez et al., 2019*).

## Histology and functions of the lung

In their description of the microanatomy of the lung, MDL start by locating the lung "between the gill, pericardial cavity [= pericardium], *foot*, and the extrapallial space that

separates the mantle and the shell'' (p. 127). Also, they indicate that the floor of the lung ''is attached to the foot'' (p. 125) and ''rests on the foot of the snail'' (p. 127). It must be pointed out that the lung is neither located near the foot nor resting on it nor attached to it, since it is situated in the roof of the mantle cavity, hanging over the cephalopodal mass (see *Rodriguez et al., 2021*).

More importantly, MDL mix up four clearly different structures: the outer mantle epithelium, the epithelial lining of the lung cavity—which they frequently and incorrectly call 'endothelium'–, the inner mantle epithelium, and the blood sinuses' walls (see figure 1 in *Rodriguez et al., 2021*). They state, for instance, that ''[a] layer of smooth muscle lies […] basal to the *endothelium* [sic, read 'epithelium'] of the *central airspace* [=lung cavity]'' (p. 128), but the two figures meant to support this statement actually depict the *wall* and *lumen* of a blood sinus and the *inner mantle epithelium* and the *mantle cavity*, respectively. In figure 6B, MDL confuse the outer mantle epithelium with the inner mantle epithelium.

Although MDL's paper deals with the microscopic anatomy of the respiratory organs in *P. maculata*, it does not include detailed micrographs of the epithelia that mediate gas exchange between blood and water (in the gill) or blood and air (in the lung). The only apparent exception is MDL's figure 8, which is supposed to show the respiratory epithelium lining the lung cavity, but instead shows the outer mantle epithelium, as is evident by the abundant pigment granules of the cells (Fig. 2A). Contrastingly, the respiratory epithelium of the lung in the related species *P. canaliculata* was recently described by *Rodriguez et al. (2021)*. This structure, which can be discerned in light micrographs (*Lutfy & Demian, 1965*; *Rodriguez et al., 2018*), constitutes the blood-air barrier in the lung and is different enough from the outer mantle epithelium so as not to confuse them (Fig. 2B).

MDL are silent about the granules that almost fill the cytoplasm of the cells in figure 8 and misidentify the rest of the structures. Specifically, they affirm that this epithelium ''is populated with numerous *calcium cells*, *nerve fibers*, and *long vertical rootlets*'' and ''contains rows of *branched and parallel cilia*'' (p. 128). But in that figure, the 'calcium cells' are actually the pigmented cells forming the outer mantle epithelium (besides, calcium cells are not epithelial cells; see *Mueck, Deaton & Lee, 2020b*). The 'long vertical rootlets' are the cell membranes of contiguous epithelial cells together with the adjacent granule-free portions of cytoplasm. The 'nerve fibers' are merely dilated intercellular spaces similar to those that have been found in the epithelium of the gill and the cellular tufts of the lung in *P. canaliculata* (*Rodriguez et al., 2019*; *Rodriguez et al., 2021*). Finally, the 'branched and parallel cilia' are actually ramified microvilli (see Fig. 2A).

Not surprisingly, the confusion between the epithelium of the lung cavity and the outer mantle epithelium leads to inconsistencies in MDL's paper. For instance, they accurately observe that the outer mantle epithelium has ''a *heavy dark pigmentation* that makes the epidermis along the roof of the lung appear black'' and whose cells ''do not bear cilia'' and that, ''[b]y contrast'', the epithelium lining the lung cavity and the inner mantle epithelium ''are *transparent*'' (p. 127) and ciliated. So, it is clear that the pigmented epithelium shown in figure 8 corresponds to the outer mantle epithelium (Fig. 2A), but they instead label it as the epithelium that lines the lung cavity (Fig. 2B). They also simultaneously maintain that the epithelial lining of the lung cavity is ''adorned with *patches* of cilia'' (p. 127)

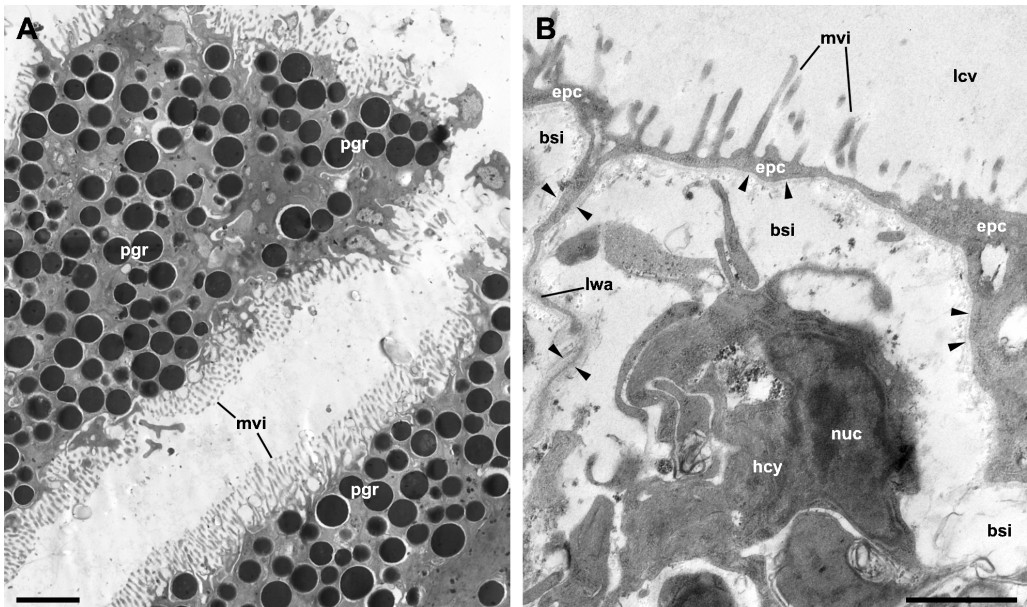

**Figure 2** **The outer mantle epithelium and the respiratory lamina in the lung of *Pomacea canaliculata*.** (A) Pigmented cells in the outer mantle epithelium. Note the similarity with MDL's figure 8. (B) The respiratory lamina lines the lung cavity and constitutes the air-blood barrier. It is composed of pavement epithelial cells and their underlying basal lamina (arrowheads) that extend as an extremely thin layer that forms the ceiling and lateral walls of a mesh of blood sinuses. Clusters of ciliated, microvillar, and secretory cells, called 'ciliary tufts', are spread in the lamina (not shown). See *Rodriguez et al. (2021)* for a thorough description of the respiratory lamina and the ciliary tufts. Abbreviations: bsi, blood sinuses; epc, epithelial cells; hcy, hemocyte; lcv, lung cavity; lwa, lateral wall of blood sinuses; mvi, microvilli; nuc, cell nucleus; pgr, pigment granules. Transmission electron microscopy. Scale bars represent: (A) 2 μm; (B) 1 μm. Unpublished micrographs courtesy of Cristian Rodriguez (Universidad Nacional de Cuyo, Argentina).

and "contains *rows* of branched and parallel cilia" (p. 128), or that "[c]iliated villi were observed *over the entire inner lining* of the lung" (p. 130). Presumably, the correct option is 'patches of cilia' since that would be consistent with more accurate descriptions in *Marisa cornuarietis* (*Lutfy & Demian, 1965*) and *P. canaliculata* (*Rodriguez et al., 2021*).

In an accompanying paper, they identify the calcium cells in the lung as rhogocytes on the basis that some discontinuities in the external lamellae of the calcium crystalloids would correspond to slit apparatuses (*Mueck, Deaton & Lee, 2020b*). But this is groundless, for slit apparatuses that characterize rhogocytes are elaborate membranous structures that connect the extracellular lacunae with the cell exterior (*Kokkinopoulou et al., 2014*; *Kokkinopoulou et al., 2015*). Besides, calcium cells in *P. maculata* do not show any of the other diagnostic features of rhogocytes, such as extracellular matrix, prominent nucleus, well-developed rough endoplasmic reticulum with dilated cisternae, and numerous granules of varying electron-density (*Haszprunar, 1996*). Granted, it has been suggested that calcium cells may be modified rhogocytes in some gastropods (see *Haszprunar, 1996* and references therein) but, from the evidence at hand in Ampullariidae, it can only be affirmed that storage cells (calcium cells in *P. maculata*; urate cells in *P. canaliculata*) and rhogocytes are two distinct

cell types that coexist in the storage tissue of the lung. *Bona fide* rhogocytes (*i.e.,* cells that show all the diagnostic features listed above) were recently showed and described for the first time in an ampullariid by *Rodriguez et al. (2021)*. Curiously, they can also be discerned in MDL's in figure 7E and in figure 4C in *Mueck, Deaton & Lee (2020b)*, although the quality and magnification of the micrographs do not allow for definitive identification. In the latter figure, a *granule* in a presumptive rhogocyte is labeled as 'mu', that is, as a *muscle cell* (this, however, can only be conjectured since the figure lacks an abbreviations list altogether). It is worth stressing that, in both papers, the authors completely overlooked these presumptive rhogocytes and thus should not be credited for these findings.

In addition to misinterpreting the morphology of *P. maculata*, MDL are imprecise regarding the functional roles of the lung in Ampullaridae. They affirm, for instance, that "ampullariids […] possess a modified mantle cavity that can act as an air sac that is used as a flotation device […]. Some apple snails in the genera *Pila* and *Pomacea* are capable of both aerial and aquatic respiration because of the retention of the gill and the development of the air sac into a vascularized lung" (p. 125). This fragment is problematic for two important reasons. First, in Ampullariidae the "air sac" (=lung) is not the modified mantle cavity as it occurs in other air-breathing gastropods, but a secondary invagination of the roof of the mantle cavity between the osphradium and the gill, which makes it unique among gastropods (see *Rodriguez et al., 2021* and references therein). Second, though the lung is indeed used as a flotation device, it is vascularized and has a respiratory function (*i.e.,* it is a true lung and not merely an air sac) presumably in all extant ampullariids species, and therefore is not exclusive of some *Pila* and *Pomacea* species. In fact, even the basally branching Old World genera *Saulea*, *Afropomus*, and *Lanistes* (*Hayes, Cowie & Thiengo, 2009*) breathe air using the lung (*Fischer & Bouvier, 1890*; *Bouvier, 1891*; *Kretschmann, 1955*; *Berthold, 1988*; *Berthold, 1991*)–with the possible exception of some *Lanistes* species that have been found at considerable depths in Lake Malawi (*Brown, 1994*).

## Final remarks

If reinterpreted in light of what we know about the functional morphology of the respiratory organs in Ampullariidae, what MDL's figures show turns out to be overall consistent with the descriptions in *P. canaliculata* by *Rodriguez et al. (2019)* and *Rodriguez et al. (2021)*. The only differences concern the "aggregations of crystals presumed to be aragonite" and cells "containing calcium inclusions" (p. 126) in the gill and the abundant calcium-storage cells in the lung (see also *Mueck, Deaton & Lee, 2020b*) of *P. maculata*, for which no equivalents were found in *P. canaliculata*. The latter may be indicative of a not well-known diversity of main metabolites accumulated in the storage tissue of the lung in different ampullariid species (the main metabolites in the storage tissue of *P. canaliculata* are urates; *Vega et al., 2007*; *Giraud-Billoud et al., 2008*). Moreover, these metabolites would be involved in estivation in *P. canaliculata* (*Giraud-Billoud et al., 2011*; *Giraud-Billoud et al., 2013*) and *P. maculata* (*Mueck, Deaton & Lee, 2020b*), which suggests the existence of alternative mechanisms of estivation and arousal that are worth exploring.

Another point that deserves more attention is the finding of endothelial-like cells in the lung's blood sinuses of both *P. maculata* (*Mueck, Deaton & Lee, 2020a*) and *P. canaliculata*

(*Rodriguez et al., 2021*). In *P. maculata*, MDL even go so far as to report a discontinuous endothelial lining (which they incorrectly call 'epithelium' in the legend of figure 6), but this is a bold claim that should be supported by more evidence.

## CONCLUSIONS

I have shown that MDL's paper features a mixture of gross morphological misinterpretations and serious conceptual and terminological errors, as well as low-quality micrographs and careless figure labeling. As a consequence, it introduces artifactual anomalous results that hardly "add to the knowledge of the anatomy of respiratory organs in molluscs" (p. 125) or, *a fortiori*, in Ampullariidae, but rather obscure the field. It may be argued that many of its most prominent mistakes could have been avoided had the authors carefully read (and discussed their results in the context of) the most detailed reports on the topic (*Lutfy & Demian, 1965*; *Rodriguez et al., 2018*; *Rodriguez et al., 2019*; see Table 1). In fact, they would have found in *Rodriguez et al. (2019)* a thorough description of the gill of *P. canaliculata* from the macroscopic to the subcellular levels, and they would have also read in *Rodriguez et al. (2018)* that the lung cavity in that species is lined by a respiratory lamina that sharply differs from the pigmented mantle epithelium. Unfortunately, these two highly relevant papers are neither discussed nor cited. However, a reinterpretation of MDL's figures shows that the morphology of the respiratory organs in *P. maculata* is similar to that of *P. canaliculata*. The only noticeable difference, which deserves further research for its implications on the mechanisms of estivation and arousal, may be the nature of the main metabolites accumulated in the lung.

## ACKNOWLEDGEMENTS

I thank Daniel S. Brooks and Alejandro Fábregas-Tejeda for commenting on the draft and Cristian Rodriguez for generously providing the micrographs in Fig. 2. All the opinions and any mistakes in this paper are entirely mine.

### Funding

The author's research is funded by the German Research Foundation (DFG; project no. BA 5808/2-1). The funders had no role in study design, data collection and analysis, decision to publish, or preparation of the manuscript.

### Grant Disclosures

The following grant information was disclosed by the author:
German Research Foundation (DFG): BA 5808/2-1.

### Competing Interests

The author declares that he has no competing interests.

## Author Contributions

- Guido I. Prieto analyzed the data, prepared figures and/or tables, authored or reviewed drafts of the paper, and approved the final draft.

## Data Availability

The raw images for Figure 2 are available as a Supplementary File.

## Supplemental Information

Supplemental information for this article can be found online at http://dx.doi.org/10.7717/peerj.12161#supplemental-information.

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
