# Peer review of "Caution ahead: reassessing the functional morphology of the respiratory organs in amphibious snails"

_PeerJ, doi:10.7717/peerj.12161_

## Round 0.1 · accepted · Accept

In the interests of transparency, you should contact the editors of the Journal of Shellfish Research and propose to link the Mueck et al. (2020) paper with your paper once it is published.

·

Basic reporting

This is a very straightforward paper criticising the findings/interpretations in another paper. The arguments are well presented and appropriately phrased, even though it is evident that the author is not happy at all about the approach taken in the criticized paper.

Experimental design

Not primary research (even though the arguments are partly based on primary research in which the author was involved), so I will not comment on the experimental design.

Validity of the findings

I am no expert on lung/gill morphology of gastropods, so I have to take the arguments at face value. Given that the author has pointed out in much detail where he finds fault with the conclusions of the criticised paper by Mueck et al. and gives a balanced view acknowledging uncertainties, I am left with the impression that the Mueck et al paper does indeed have quite a few flaws. In the very least, this will inspire readers to draw their own conclusions in light of the arguments made in this MS, so I think it is worthy of publication even if there should be scope for discussion about some of the points made. The only thing I am wondering about is why this is not published in the same journal as the criticised paper - has this been tried? Surely an editor must take a criticism of this scope into account (and offer a reply to the authors of the criticised paper).

Additional comments

Line 162: I think "do not show" is missing before "any"? The sentence would make more sense then.

Reviewer 2 ·

Basic reporting

This is an unusual paper in that it is mainly a critique of (two) recent publications (Mueck, Deaton & Lee 2020a,b: J Shellfish Res 39(1): 125-132, 133-141) rather than an original contribution. It is obvious (and I agree with the details) that the criticized papers do have a number of shortcomings (confusions and misinterpretations), nevertheless:

(1) This critique should preferably be published in the original journal of the criticized papers (Journal of Shellfish Research) and

(2) In any case the criticized authors (Mueck, Deaton & Lee) should be given the possibility to answer the critique, and their response should be published side by side with this critique in order to be fair.

Experimental design

no comment

Validity of the findings

see above under basics

·

Basic reporting

This is a well written and justified critique of Mueck et al. 2020. The author does a nice job of pointing out major errors and misinterpretation in Mueck et al. 2020 and the implications such issues will create for interpretation and understanding of the biology, evolution, and functional morphology of ampullariids. Given the large number of errors and the ease at which they could have been caught in peer review, this should never have made it to press. Unfortunately, this happens, which in turn necessitates corrective reviews such as this one. Hopefully Mueck et al. 2020 and the editors of the editors of the Journal of Shellfish Research will provide a link to this paper once it is published so others less familiar with ampullariid anatomy won't be misled and the errors compounded.

I only noted a few minor spelling and grammatical errors, but nothing of substance that would necessitate revisions.

Experimental design

There were no experiments, as this was a critique of a previously published study. The author does a nice job of laying out the issues and references them very well.

Validity of the findings

The underlying premise of the critique is robust and fully supported with clear examples and solid citations. The conclusions are wells stated and limited to the points raised throughout.